# Local Predictors of Explosive Synchronization with Ordinal Methods

**DOI:** 10.3390/e27020113

**Published:** 2025-01-24

**Authors:** I. Leyva, Juan A. Almendral, Christophe Letellier, Irene Sendiña-Nadal

**Affiliations:** 1Complex Systems Group & GISC, Universidad Rey Juan Carlos, 28933 Móstoles, Spain; inmaculada.leyva@urjc.es (I.L.); juan.almendral@urjc.es (J.A.A.); 2Center for Biomedical Technology, Universidad Politécnica de Madrid, 28223 Pozuelo de Alarcón, Spain; 3Campus Universitaire du Madrillet, Rouen Normandie Université-CORIA, F-76800 Saint-Etienne du Rouvray, France; christophe.letellier@coria.fr

**Keywords:** ordinal patterns, ordinal permutation entropy, ordinal transition network, chaotic synchronization, complex networks, early warning signals

## Abstract

We propose using the ordinal pattern transition (OPT) entropy measured at sentinel central nodes as a potential predictor of explosive transitions to synchronization in networks of various dynamical systems with increasing complexity. Our results demonstrate that the OPT entropic measure surpasses traditional early warning signal (EWS) measures and could be valuable to the tools available for predicting critical transitions. In particular, we investigate networks of diffusively coupled phase oscillators and chaotic Rössler systems. As maps, we consider a neural network of Chialvo maps coupled in star and scale-free configurations. Furthermore, we apply this measure to time series data obtained from a network of electronic circuits operating in the chaotic regime.

## 1. Introduction

Critical transitions, or tipping points, refer to sudden and often irreversible changes in the behavior of systems occurring in various natural and engineered contexts. These phenomena are especially concerning in fields such as ecology [1,2], neuroscience [3,4,5], climate science [6,7,8], and even financial markets, where predicting such transitions is crucial for mitigating adverse outcomes [9,10]. Researchers have extensively studied early warning signals (EWSs) to identify impending transitions [11,12]. These signals include critical slowing down [13,14,15,16], intermittency [17,18], and pattern formation [15,19], which appear in many systems.

Complex networks, which describe interconnected systems such as ecosystems, brain networks, and social interactions, are particularly challenging for transition prediction. In these systems, critical transitions often manifest as changes in the collective synchronization state. The shift to a coherent state can enhance the functionality of networked systems, as seen in brain networks [20] and power grids [21]. However, this transition to synchronization can also have catastrophic consequences, such as epileptic seizures [3,5], fibromyalgia [4], or species extinction [22,23]. Understanding how network topology influences system dynamics is essential for predicting synchronization transitions, and several studies have shown that features of the structural [24,25] or functional [26] network can serve as indicators for the transition. Identification of sentinel nodes, displaying EWSs more prominently than others, has significantly improved prediction accuracy [22,27]. Sentinel nodes have been utilized in ecological networks to detect potential regime shifts and in brain networks to pinpoint critical regions for neuromodulation [4,24].

However, traditional EWS measures such as autocorrelation, fluctuation variability, or kurtosis are not universally reliable. In highly nonlinear or stochastic systems, their effectiveness is significantly diminished. For instance, as systems approach critical transitions, higher-order nonlinear effects, often overlooked in linear approximations, become increasingly important [11,12,28,29,30]. Additionally, traditional methods may struggle to distinguish meaningful signals from noise in systems with high-dimensional dynamics, emphasizing the need for more robust techniques [25,27,31].

Recently, machine learning (ML) methods have demonstrated the potential to address challenges by identifying complex patterns in data that conventional EWSs may overlook. Studies have demonstrated the utility of ML in predicting tipping points in oscillator ensembles [10,32,33,34], ecology systems [23,35] or brain networks [32,36]. Additionally, ordinal methods, which analyze the relative order of data, have emerged as a promising tool for predicting transitions [19,37,38]. By focusing on the sequence of events rather than absolute values, ordinal analysis proves to be robust against noise and non-stationarity, making it well suited for detecting subtle precursors to changes in evolving systems [39,40,41,42].

A further challenge in prediction is the occurrence of explosive transitions. This term encompasses various phenomena, such as explosive synchronization (ES) [43], sudden structural changes in adaptive systems [44], and abrupt shifts in social dynamics [45,46]. Explosive transitions have profound implications for real-world systems, as they have been linked to conditions such as epilepsy [47], hypersensitivity in brain networks [4], and massive extinctions [35]. Predicting these sudden transitions is particularly challenging because collective observables tend to remain stable until the moment of transition. Traditional EWSs often struggle to predict explosive synchronization due to its rapid onset [48]. In addition, EWSs are studied and designed for forecasting transitions mainly in stochastic systems [11]. For systems embedded in a weakly coupled network, the inputs from their neighborhood act as an incoherent source in each node’s dynamics, playing a perturbation role, but cannot be considered a stochastic source. Although machine learning approaches have proven helpful in detecting explosive synchronization in small ensembles [23,35], there is still a need for alternative methods that are computationally less demanding.

Recently, ordinal analysis applied to time series from single nodes of dynamical networks [40,49] has revealed a strong correlation between ordinal pattern transition (OPT) entropy and structural and dynamical centrality in a wide range of weakly coupled heterogeneous ensembles. This suggests that ordinal local measures, particularly OPT, can effectively rank the sensitivity of nodes based on their proximity to transitions [25,27]. This work explores the ordinal transition entropy measured at sentinel central nodes as a potential predictor of explosive transitions to synchronization. Our findings indicate that the OPT entropy adds more information to traditional EWS measures and could be a valuable addition to the tools available to predict critical transitions.

## 2. Methods

### 2.1. Ordinal Patterns and Permutation Entropy Measures

Ordinal pattern-based methods are a family of simple and robust techniques that characterize time series simply by comparing neighbouring relative values [39,50,51,52]. Given any time series $x=\{x_{t};t=1,\ldots ,T\}$, data are projected into a sequence of symbols of ordinal patterns of a given length D>1 obtained from the comparison of consecutive (τ=1) or non-consecutive (τ>1) points in the following manner. First, the time series is divided into blocks $v_{t}=\left(x_{t},x_{t+\tau },\ldots ,x_{t+(D-1)\tau }\right)$ of size *D*, and then each element of the block is replaced by a number in [1,…,D] corresponding to its relative position when arranged in ascending order. In this way, each block $v_{t}$ is assigned to one of the D! possible orderings (permutations) $\pi_{\ell }$ in which the *D* elements can be ordered. Finally, the probability at which the ordinal pattern $\pi_{\ell }$ is found in the time series is computed as $p\left(\pi_{\ell }\right)=\#\left\{S_{t}\;\mathrm{is}\;\mathrm{mapped}\;\mathrm{into}\;\mathrm{symbol}\;\pi_{\ell }\right\}/L$, with L=⌊T/(τD)⌋ being the total number of blocks $S_{t}$ in which the time series is divided (⌊⌋ is the floor function). This procedure enables a reliable symbolic dynamics mapping a time series to a symbolic sequence of ordinal patterns (where each sequence of *D* integers receives a specific symbol). The time series has to be sufficiently long, L≫D!, to obtain reliable statistics and a valuable ordinal pattern probability distribution $P=\{p\left(\pi_{\ell }\right),\ell =1,\ldots ,D!\}$ [50]. Throughout this work, we will use D=3 (Bandt and Pompe suggest using 3≤D≤7 for practical purposes [50]) and T=2000. Although the time lag τ at which the data are sampled is not critical, this parameter could be relevant when the system under study has intrinsic time scales [53,54]. Typically, to overcome the dependency of the ordinal pattern probability distribution on the sampling time τ, its selection is usually based on time delay embedding criteria [51,55]. In addition to using a fix sampling time applied to the original time series, we will also consider a Poincaré section approach by retaining the relative maxima of the signal [56,57].

Our goal is to determine whether the probability distribution of ordinal patterns associated with the time series of a dynamical network is able to detect hidden clues in its collective state, pointing to the occurrence of an eventual abrupt phase transition. To achieve this, we will compare two entropic quantities estimated from the sequence of the permutation patterns defined above: the permutation entropy and the transition permutation entropy.

#### 2.1.1. Ordinal Permutation Entropy

Given the *D*-order permutation probability distribution $P=(p\left(\pi_{1}\right),\ldots ,p\left(\pi_{D!}\right))$, the Bandt–Pompe permutation entropy is the corresponding Shannon entropy evaluated as(1)$$S\left[P\right]=-\sum_{\ell =1}^{D!}p_{\ell }\ln p_{\ell },$$
with the criterion $0^{0}=1$. We define a normalized permutation entropy as(2)$$H\left[P\right]=\frac{S}{S_{\max}}$$
where $S_{\max}$ is the Shannon entropy of the uniform permutation probability distribution—that is, $S_{\max}=S\left[P_{e}\right]$ with $P_{e}=\left(\frac{1}{D!},\ldots ,\frac{1}{D!}\right)$.

#### 2.1.2. Ordinal Permutation Transition Entropy

While the previous entropy measure is solely based on the appearance of permutation patterns, transitions between patterns may reveal information about the finer temporal organization of a dynamical system [58]. We define the ordinal transition probability matrix $O_{T}:=\left(p_{\ell m}\right)$ as(3)$$p_{\ell m}=\frac{\#(\pi_{\ell },\pi_{m})}{\#(\pi_{\ell })}$$
with $p_{\ell m}$ being the probability that pattern *m* follows pattern *ℓ*. In case $\#(\pi_{\ell })=0$ for some $\pi_{\ell }$, we assume $p_{\ell m}=0$. We move from having D! patterns to $D{!}^{2}$ transitions. Thus, the time series must be longer to make the transition matrix $O_{T}$ statistically significant.

Since $\sum_{m}p_{\ell m}=1$, $O_{T}$ is a column-stochastic matrix (weighted and directed) whose coefficients are the transition probabilities among ordinal patterns, including self-transitions. Thus, we can characterize the dynamics of the transition patterns at the local and global level of a time series. At the local level, we can define the entropy of each pattern $\pi_{\ell }$ through the distribution probability of the patterns that follow it as(4)$$H_{\pi_{\ell }}=-\frac{1}{lnD!}\sum_{m_{=}1}^{D!}p_{\ell m}\ln p_{\ell m}.$$
which quantifies the predictability of the local transitions from the ordinal pattern $\pi_{\ell }$ to any other pattern [59,60].

At the global level, we measure the transitional complexity of the whole $O_{T}$ as the average of the local (pattern) entropies $H_{\pi_{\ell }}$:(5)$$H_{T}=\frac{1}{D!}\sum_{\ell =1}^{D!}H_{\pi_{\ell }}$$

### 2.2. Dynamical Networks and Local Early Warning Indicators

We consider networks whose nodes are either continuous (a flow) or discrete (a map) dynamical systems on the phase space. In particular, as an example of flows, we will investigate networks of *N* diffusively coupled phase oscillators and chaotic Rössler systems, while for maps, we consider a neural network of Chialvo maps. The general form of the equations governing the dynamics in each case are, respectively,(6)$${\dot{x}}_{i}=F_{i}\left(x_{i}\right)+d\sum_{j=1}^{N}a_{ij}H\left(x_{j}-x_{i}\right)\;,$$
where $x_{i}\left(t\right)$ is the vector state of node *i* at time *t* and $F_{i}$ is the local vector field, and for the map,(7)$$x_{i}\left(t+1\right)=M_{i}\left(x_{i}\left(t\right)\right)+d\sum_{j=1}^{N}a_{ij}H\left(x_{j}\left(t\right)-x_{i}\left(t\right)\right)\;,$$
where $x_{i}\left(t\right)$ is a *m*-dimensional vector describing the state of each node *i* at the *t*-th iteration of the map $M_{i}$. In both cases, H is the output function describing the units’ interaction. The coupling architecture among them is defined by the adjacency matrix *A* of the graph whose coefficients $A:=(a_{ij})$ are $a_{ij}=1$, if *i* and *j* are connected, and $a_{ij}=0$ otherwise, such that, the degree of each node $k_{i}=\sum_{j}a_{ij}$. The parameter $d=\sigma /k_{\max}$ acts as a normalized coupling strength by the maximum degree of the network $k_{\max}=\max\left(k_{i}\right)$ to compare different realizations of the network.

Depending on the local functions F and M, the output function H, and on the adjacency matrix *A*, the dynamical networks described by Equations (Equation 6) and (Equation 7) may display a transition from a non-synchronous state to a synchronized state when the strength of the coupling *d* is above a critical value [61]. To monitor the collective state of the network for increasing values of the control order parameter *d*, we compute the time-averaged stationary value of the phase order parameter,(8)$$R=\frac{1}{N}{\langle |\sum_{j=1}^{N}e^{i\theta_{j}\left(t\right)}|\rangle }_{t}$$
where $\theta_{j}$ is the associated phase of the oscillator whose state is $x_{j}\left(t\right)$. The phase-order parameter *R* accounts for the level of phase synchronization (0≤R≤1), and ${\langle \rangle }_{t}$ stands for the time average along a sufficiently large time series once a stationary state is reached. In general, we compute two synchronization curves for the phase order parameter *R*, the forward and backward continuations. The forward curve is performed by gradually increasing the value of *d*, starting from random initial conditions, and computing the stationary value of *R*, and then taking the final stationary state as the initial state for the next value of *d*. For the backward curve, *d* is gradually decreased starting from initial conditions close to a stable synchronous state and the network eventually returns to a non synchronous state when d=0.

Transitions to synchronization can be abrupt, also called explosive transitions, when the order parameter remains R≃0 as the coupling *d* increases until suddenly it jumps to R≃1 and the network stays in a synchronized state. They often exhibit hysteresis with the backward transition occurring at lower values of the coupling strength. Explosive phase synchronization appears when the interaction between the network structure and the local node dynamics hinder the formation of microscopic seeds of synchronization as the coupling increases, a mechanism underlying the emergence of a collective coherent state in smooth, second-order transitions. As a result, the system remains incoherent despite very high coupling strengths, only to abruptly transition to a fully synchronized state [43].

Most practical ways to create this effect are based on inducing *frequency dissasortativity*, ensuring that each node is dynamically isolated [62]. When the network is structurally heterogeneous, this dynamical separation is easily achieved by imposing a degree–frequency correlation [63,64]. Therefore, in this work, we study highly heterogeneous topologies (star and scale-free (SF) networks), where the nodes are distributed such that their inner temporal scale is proportional to their degree. In the case of the star configuration, this rule is slightly modified by adding some randomness to the leaves, that is, nodes with $k_{i}=1$. This quenched disorder is drawn from a uniform distribution U(0,ξ) and avoids a persistent synchronous state when reducing the coupling strength in the backward route to desynchronization. This parametric uncertainty is very small, producing slight variations of the critical coupling strength for the ES when performing different simulation instances, and preserves the frequency gap between the hub and the leaves; otherwise, the transition to synchronization is no longer abrupt. Other sources of stochasticity come from the unavoidable tolerances on the electronic component parameters and the noise from uncontrolled sources in the experimental setup that will be described in Section 3.4. Details of the dynamical ranges used in each system are provided in their respective sections.

As advanced in Section 2.1, we propose using a local entropic measure as an EWS of an abrupt transition in a dynamic network. For each node *i*, we consider one of the accessible scalar variables of the vector state $x_{i}\left(t\right)$, or, when convenient, a function of that variable, and compute the corresponding permutation transition entropy $H_{T}^{i}$. We expect that nodes with the same degree *k* will have similar dynamical patterns within the network [49], allowing us to define the averaged k−class permutation transition entropy(9)$${\langle H_{T}\rangle }_{k}=\frac{1}{N_{k}}\sum_{i|k_{i}=k}H_{T}^{i},$$
where $N_{k}$ is the number of nodes having degree *k*, and ${\langle \rangle }_{k}$ is an ensemble average. Similarly, we define a *k*-class average for the permutation entropies of those nodes with the same degree ${\langle H\rangle }_{k}$.

Finally, to compare the performance of the proposed local entropic measure given by Equation (Equation 9) with other common early warning indicators for critical transitions used in the literature [1], in some cases, we will monitor the amplitude of the fluctuations of each node time series $x_{i}\left(t\right)$, $\sigma_{f,i}=\sqrt{\langle x_{i}{\left(t\right)}^{2}\rangle -{\langle x_{i}\left(t\right)\rangle }^{2}}$, and the normalized autocorrelation(10)$$AC_{i}\left(l\right)=\frac{\sum_{t=1}^{T-l}x_{i}\left(t\right)\cdot x_{i}\left(t+l\right)}{\sqrt{\sum_{t=1}^{T-l}x_{i}^{2}\left(t\right)\cdot \sum_{t=1}^{T-l}x_{i}^{2}\left(t+l\right)}}$$
with lag l=1.

## 3. Results

In the following sections, we present results demonstrating that the transition entropy measured at the hubs of a network is a highly effective metric anticipating the onset of abrupt synchronization transitions in networks of various dynamical systems with increasing complexity. These systems include phase oscillators, a map emulating neuronal firing dynamics, and chaotic amplitude oscillators. Additionally, we apply this measure to experimental time series data obtained from a network of electronic circuits.

### 3.1. Abrupt Synchronization Transition in Kuramoto Phase Oscillators

The first studied system is the Kuramoto network,(11)$${\dot{\theta }}_{i}=\omega_{o,i}+d\sum_{j=1}^{N}a_{ij}\sin\left(\theta_{j}-\theta_{i}\right)$$
where the coupling *d* is the coupling strength and $\omega_{o,i}$ is the natural frequency of node *i*. In the example in Figure 1, we use a *N* = 31 star-like structure, and to achieve explosive synchronization, we impose a frequency–degree correlation in the nodes [43,63], with $\omega_{o,h}=1.3$ for the hub and $\omega_{o,l}=1+0.005\epsilon$ for the leaves with ϵ a [0,1].

When searching for signals that indicate an imminent transition, one important decision is determining which observable to monitor, as not all the system outputs are equally sensitive to changes in the state [13,24,65,66]. In networked systems, not all nodes are equally effective as predictors of transitions, and their sensitivities may vary due to internal dynamics or their position within the network structure [10,27]. In a Kuramoto network, using the global instantaneous synchronization state R(t) as an EWS could provide moderate utility [48], but this requires access to a global variable that may not be readily available. In our case, we are more interested in local measures. Therefore, in Figure 1, we use the instantaneous frequency of each oscillator $\omega_{i}\left(t\right)$, which is recorded with a sampling time of τ=200 integration steps.

As seen in ES transitions, the average order parameter *R* does not provide information about the transition and remains close to its initial value until it suddenly jumps to its maximum. We then evaluate the ordinal permutation entropy ${\langle H\rangle }_{k}$ and the ordinal permutation transition entropy ${\langle H_{T}\rangle }_{k}$ at both $k_{\mathrm{hub}}=N-1$ for the hub and $k_{\mathrm{leaf}}=1$ for the leaves [Figure 1a]. For the leaves, the entropy measurements show minor sensitivity to the proximity of the transition. However, at the hub, both entropies increase noticeably when the coupling is still one-third of its critical synchronization value. Notably, the relative increase in $H_{T}$ is much more significant than that of H, providing a clear advantage for generating an early detection alarm. Given this advantage, the remaining sections will focus on exploring $H_{T}$ as a more promising measure. Specifically, we propose as EWS measure the increase in the relative difference between the OPT entropies of the most connected and less connected nodes in the network.

We now compare the result of the local entropy measures with some of those traditionally used as early warning predictors (Figure 1), the standard deviation of the fluctuations $\sigma_{f,i}$ and the 1-lag autocorrelation AC_i_(1) of the same data analyzed in panel (a). As for the entropies, for the leaf, none of these observables present significant variations that can be considered EWSs. However, the dispersion of fluctuations exhibits a linear growth proportional to the coupling that results from the increase in the amplitude of the frequency beat, before sharply dropping simultaneously to the transition, recovering the value for the uncoupled system d=0 as expected. The AC(1) measure for the hub produces a result comparable to H.

On the other hand, the fluctuation dispersion $\sigma_{f}$ of the hub shows a relatively significant maximum (note that the scale of this measure refers to the right Y-axis). However, this maximum does not occur near the synchronous forward transition we are studying, but rather in the region associated with the critical coupling for *desynchronization* in the backward transition, a much smaller coupling due to the wide hysteresis. This interesting effect would, however, lead to a false alarm regarding the ES transition.

### 3.2. Abrupt Synchronization Transitions in Coupled Chialvo Maps

To illustrate the performance of the ordinal transition entropy to foresee an upcoming network synchronization event in networks of coupled maps, we consider a neural network of Chialvo maps. The Chialvo map is a two-dimensional neural model that produces periodic or chaotic burst-spike dynamics [67], and it has the following form:(12)$$\begin{array}{cc} x_{i}\left(t+1\right) & =x_{i}^{2}\left(t\right)\exp\left(y_{i}\left(t\right)-x_{i}\left(t\right)\right)+I_{i}+d\sum_{j=1}^{N}a_{ij}\left(x_{j}\left(t\right)-x_{i}\left(t\right)\right),\\ y_{i}\left(t+1\right) & =ay_{i}\left(t\right)-bx_{i}\left(t\right)+c \end{array}$$
where $x_{i}\left(t\right)$ acts as a membrane potential and $y_{i}\left(t\right)$ as a restoring variable. The subscripts *t* represent a discretized time evolution (t=1,…,T), while subscript *i* refers to a node (neuron) of the network (i=1,…,N). Parameters a=0.89, b=0.6, and c=0.28 are chosen such that the neural dynamics are periodic or quasi-periodic, whose frequency and amplitude is modulated by the parameter *I*, which plays the role of an ion current injected into the neuron. As *I* increases, the frequency increases monotonically [67] (an extensive study of the parameter space of this map is reported in Ref. [68]). Here, we impose a linear dependence of the constant bias parameter *I* with the neuron connectivity as $I_{i}=0.049+\alpha k_{i}$ to induce an explosive synchronization when increasing the control order parameter *d*.

Figure 2a,b shows an example of the time evolution of the activation variable $x_{t}$ and phase portrait $\left(x_{t},y_{t}\right)$ of an isolated neuron for $I_{i}=0.05$. The red segments connect the spike maxima with $x_{t}>0.5$, used to define a phase $\theta_{i}\left(t\right)$ and be able to monitor the network synchronization as described in Section 2.2. The phase increases by 2π each time there is a spike, and it is linearly interpolated between spikes [69] as(13)$$\theta_{i}\left(t\right)=2\pi n_{i}+2\pi \frac{t-t_{i,n}}{t_{i,n+1}-t_{i,n}},\;t_{i,n}⩽t<t_{i,n+1}$$
where $t_{i,n}$ is the time at which the *i*-th neuron fires its $n_{i}$-th spike.

The sequence of maxima is used to construct the ordinal patterns and compute the ordinal transition entropy of Chialvo maps coupled in star and scale-free configurations. Figure 2 shows the evolution of the *k*-class transition entropy ${\langle H_{T}\rangle }_{k}$ for nodes of increasing degree (colored curves) belonging to a star of size N=31 (panel c) and to a scale-free network of size N=100 (panel d) as the coupling *d* increases. In a star configuration, there are only two classes of sensors: the hub and the leaves. In contrast, a scale-free network exhibits a more diverse degree distribution, where many nodes have lower degrees while only a few have significantly larger degrees. The nodes with higher degrees are particularly good candidates for signaling an abrupt phase transition. As in the Kuramoto case, we observe how, in the two configurations, the largest degree class nodes exhibit a pronounced growth of their transition entropy long before the sudden jump in the order phase parameter *R* (black continuous line), while the lowest degree ones do not provide information about the abrupt change occurring in the network phase synchronization.

### 3.3. Predicting Explosive Transitions in Networks of Rössler Oscillators

We now consider a dynamical network composed of chaotic oscillators, adding an extra level of complexity to the local dynamics. We chose Rössler systems [70] coupled through the *y* variable, whose governing equations are the following:(14)$$\begin{array}{cc} {\dot{x}}_{i} & =-w_{i}y_{i}-z_{i},\\ {\dot{y}}_{i} & =w_{i}x_{i}+ay_{i}+d\sum_{j=1}^{N}a_{ij}\left(y_{j}-y_{i}\right),\\ {\dot{z}}_{i} & =b+z_{i}\left(x_{i}-c\right), \end{array}$$
where a=0.165, b=0.4, and c=8.5 are set to obtain a phase-coherent chaotic attractor, and $w_{i}$ is a control parameter that directly tunes the main frequency of the chaotic oscillator. We set the linear frequency–degree correlation as $\omega_{i}\left(k\right)=1.06+\alpha k_{i}$ with $\alpha =2.73\times 10^{-4}$ to induce an explosive synchronization transition. To monitor the network state of phase synchronization, we define the phase of each Rössler oscillator as $\theta_{i}=\arctan\left(y_{i}/x_{i}\right)$ and compute the phase order parameter as given by Equation (Equation 8).

To investigate this dynamical network, we chose to map each node’s vector state $x_{i}=\left(x_{i},y_{i},z_{i}\right)$ to the one-dimensional time series $y_{i}\left(t_{m}\right),m=1,\ldots ,T$ generated by the Poincaré section $P\equiv \left\{\left[x_{i}\left(t_{m}\right),z_{i}\left(t_{m}\right)\right]\in R^{2}|{\dot{y}}_{i}\left(t_{m}\right)=0,{\ddot{y}}_{i}\left(t_{m}\right)>0\right\}\;$ [37]—that is, we take the minima of the *y* time series to build the D=3 permutation patterns.

Figure 3 condenses the main results, evidencing again how the hubs’ transition entropy $H_{T}$ detects an explosive synchronization event in advance while the phase order parameter *R* does not sense any collective effect. In detail, Figure 3a shows a markedly distinct behavior of the OPT entropies for the leaf and hub of a star network. While the $\langle H_{T}\rangle$ of the leaf remains almost flat along the synchronization path, the hub’s entropy rises at coupling values even before the hysteresis window starts, defined by the continuous and dashed *R* curves for increasing ($R_{\mathrm{for}}$) and decreasing ($R_{\mathrm{back}}$) strength of the coupling. In more complex situations with a heterogeneous degree distribution, as shown in Figure 3b, we observe a similar hierarchical trend where nodes in higher-degree classes begin to gain entropy at smaller coupling values. Figure 3c shows the sensitivity of the OPT entropy as a function of the node’s degree depending on the distance from the abrupt transition. While for very low coupling values ($d=0.08\times 10^{-3}$), the ${\langle H_{T}\rangle }_{k}$ monotonously decreases with *k*, for values closer to the hysteresis window ($d=0.6\times 10^{-3}$), there is a clear cutoff around $k_{c}∼50$ indicating that nodes with $k>k_{c}$ are potential EWS sensors.

### 3.4. Predicting Explosive Transitions in Networks of Electronic Circuits

Finally, we test the performance of the OPT entropy $H_{T}$ as an early warning measure for an experimental case of explosive synchronization. The experiment consists of an N=6 star network of piecewise Rössler electronic circuits operating in the chaotic regime, as the numerical counterpart in the previous section. The circuits are configured such that the central node oscillates with a mean frequency of 3333 Hz, and the leaf nodes are set with frequencies in the range of 2240 ± 200 Hz. The transition to synchrony of this ensemble was first analyzed in Ref. [64], where the rest of the experimental details can be found and the dataset can be downloaded from [71].

In the numerical analysis presented in Section 3.3, the node coupling and the variable being analyzed correspond to $y_{i}$, while in the experiment, this role is assigned to *x*. To align our analysis with the results above, we extract the Poincarè section from the experimental data to generate the D=3 permutation patterns. An experimental parameter controls the node’s dynamics so that transition can be tuned from second-order (Figure 4a) to explosive (Figure 4b) without affecting the node’s frequencies or the chaotic state. The phase reconstruction and synchronization monitoring are performed as in Section 3.3.

For a better comparison, in Figure 4, we normalize the OPT entropies of each node ${\langle H_{T}\rangle }_{k}$ to their respective value when the system is uncoupled, ${\langle H{\left(0\right)}_{T}\rangle }_{k}$. In the continuous transition (a), neither the hub nor the leaves show changes in their initial entropy value along the synchronization process. In contrast, during the ES transition plotted in Figure 4b, the hub OPT entropy increases by up to a factor of five when the coupling value is still far from the transition. Meanwhile, the leaf experiences a slight reduction before the transition. This result is similar to the one obtained in Figure 3a for the numerical case, which confirms the robustness of this ordinal method as an EWS for the explosive transition even in this case subject to the node’s experimental heterogeneity and noisy environment.

## 4. Conclusions

Predicting state transitions in complex systems is a difficult challenge due to the enormous variety of systems, forms of transition, and diversity of data. In the case of abrupt transitions, it is even more difficult, since many of the system’s observables do not show changes during the stages before the transition. In this work, we explore the efficacy of ordinal pattern transition (OPT) entropy as a predictive tool for explosive synchronization transitions in various dynamical networks. Through simulations and experimental validations, OPT entropy can identify subtle signals of the proximity of the transitions across diverse network configurations and dynamical regimes, even outperforming traditional EWSs.

Our approach differs from previous studies because the observable used as input—such as the instantaneous frequencies of individual nodes in the case of the Kuramoto model, or the local Poincaré sections for the Rössler and Chialvo maps—is not the same as the one that undergoes ES, which is the global phase synchronization value *R*. In extended networked systems, it is more common to have access to local measurements rather than global synchronization data.

Our results indicate that the sensitivity of OPT entropy is particularly strong in central nodes, which can function as sentinel nodes based on this observable measure. We successfully applied this method to chaotic networks of electronic circuits, demonstrating its versatility and practical utility in real-world applications. One promising application is the prediction of epileptic seizures, yet the performance of permutation entropy from electroencephalogram recordings still remains unclear [72,73,74,75]. To our knowledge, OPT entropy has not yet been explored to analyze the transition process from normal to seizure state in spatially distributed EEG signals. The combination of a complexity measure that captures subtle changes in the temporal organization of a time series while considering the underlying network structure of the EEG signals deserves attention and further investigation. These findings highlight the potential to incorporate ordinal methods into the toolkit for studying critical transitions in complex systems, especially when traditional EWS measures struggle under high dimensionality or non-linearity conditions.

Future research could focus on refining ordinal measures to enhance their sensitivity and applicability to larger and more diverse networks or to analyze the influence in the prediction of different types of observational and dynamical noise [11,12]. Additionally, integrating machine learning with ordinal methods may prove beneficial in extending their predictive capabilities in various fields such as neuroscience, ecology, and engineering. This study paves the way for broader applications of ordinal methodologies in forecasting and mitigating the effects of critical transitions in complex dynamical systems.

## Figures and Tables

**Figure 1 entropy-27-00113-f001:**
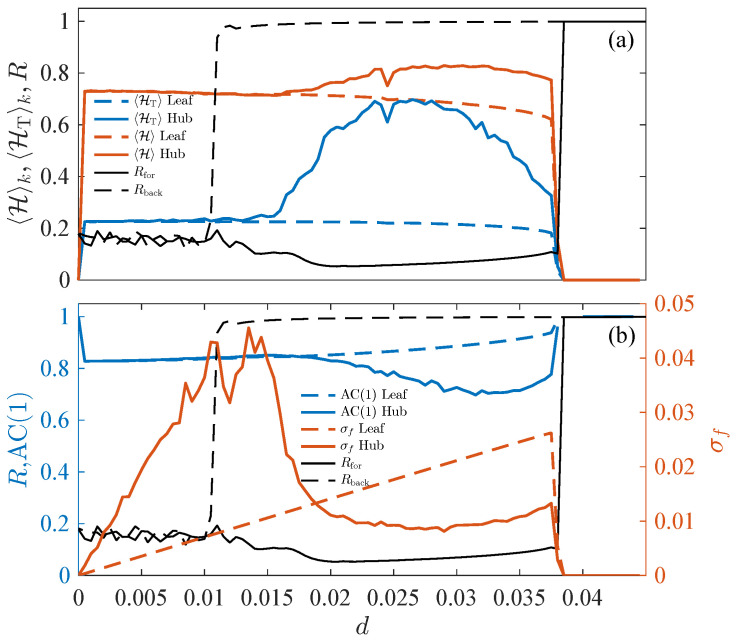
Abrupt synchronization transition in Kuramoto phase oscillators. (**a**) ${\langle H\rangle }_{k}$ and ${\langle H_{T}\rangle }_{k}$ as a function of the normalized coupling d=σ/(N−1) in a Kuramoto star network of size N=31 with $\omega_{o,h}=1.3$ for the hub and $\omega_{o,l}$ = 1 + 0.005ϵ for the leaves with ϵ a [0,1] random number. Black lines correspond to the Kuramoto order parameter *R* for the forward (continuous lines) and the backward (dashed lines) transitions. (**b**) Normalized 1-Lag autocorrelation AC(1) (blue lines, left y-axis for scale) and standard deviation $\sigma_{f}$ (orange lines, right y-axis for scale) of the signal fluctuations. Input data are a τ=200 periodic sampling of the instantaneous frequency $\omega_{i}\left(t\right)$ series. Results averaged over 10 random instances.

**Figure 2 entropy-27-00113-f002:**
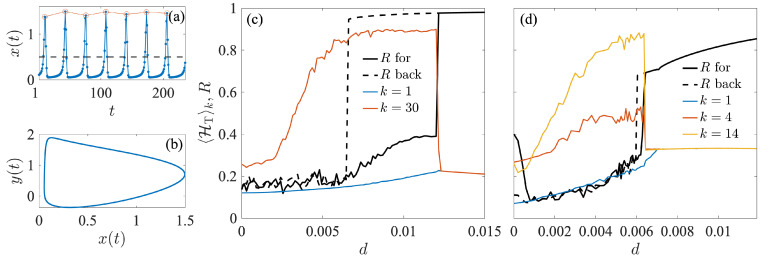
Anticipating abrupt synchronization in networks of coupled Chialvo maps. (**a**) Time series for $x_{t}$ and (**b**) phase portrait $x_{t}-y_{t}$ of an isolated neural model (Equation 12) for a=0.89, b=0.6, c=0.28, and I=0.05. Red segments in (**a**) connect spike maxima with $x_{t}>0.5$. (**c**,**d**) ${\langle H_{T}\rangle }_{k}$ for several node degrees (see legend) and Kuramoto order parameter *R* as a function of the coupling strength *d* for (**c**) a star graph of size N=31 with I=0.050 for the hub and $I=0.049+10^{-4}\epsilon$ for the leaves, with ϵ, a random number uniformly distributed in [0,1], and (**d**) a scale-free network of N=100 nodes of mean degree 4, with I(k)=0.049+αk and $\alpha =3\times 10^{-5}$. In each case, there is an abrupt transition to phase synchronization (black solid line) and an abrupt transition back to desynchronization (black dashed line) with a large hysteresis behavior for the star configuration.

**Figure 3 entropy-27-00113-f003:**
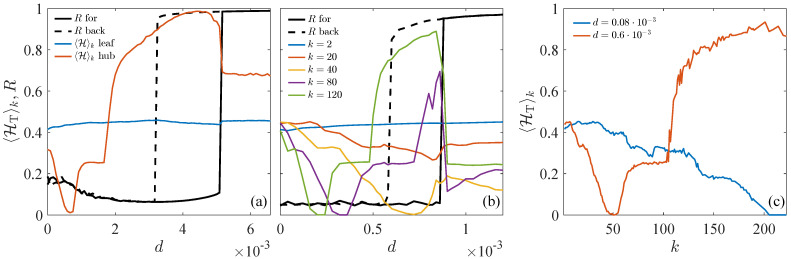
Predicting explosive transitions in networks of Rössler oscillators. (**a**,**b**) ${\langle H_{T}\rangle }_{k}$ for different *k* values (colored lines) and order parameter *R* for the forward (black continuous line) and backward (black dashed line) routes to explosive synchronization/desynchronization as the coupling strength *d* increases/decreases for (**a**) a star of size N=31 and (**b**) a N=500 scale-free network (〈k〉=4, γ=2.25). (**c**) ${\langle H_{T}\rangle }_{k}$ is a function of *k* for two values of the coupling strength in (**b**): a relatively small coupling value, $d=0.08\times 10^{-3}$ (blue line), and a value still far from the transition, $d=0.6\times 10^{-3}\ll d_{t}∼0.9\times 10^{-3}$ (red line). In (**b**,**c**), each point averages the result of 64 independent network realizations.

**Figure 4 entropy-27-00113-f004:**
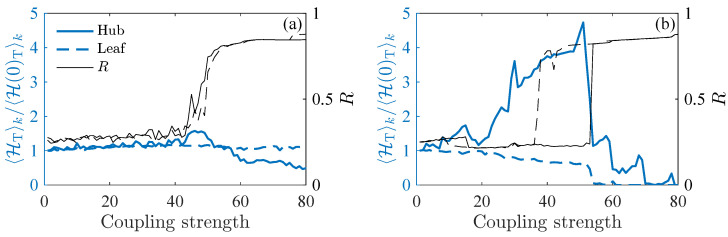
Normalized ordinal pattern transition entropies ${\langle H_{T}\rangle }_{k}/{\langle H{\left(0\right)}_{T}\rangle }_{k}$ (left y-axis) and phase order parameter *R* (right y-axis) as a function of the coupling strength in a star network of N=6 piecewise Rössler electronic circuits for the case of (**a**) continuous and (**b**) explosive transitions to synchronization. Input data are Poincaré sections series of a voltage local maxima.

## Data Availability

The experimental dataset used in this study is available at https://doi.org/10.5281/zenodo.14146078 (see Ref. [71]).

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
