# Peer review of "Local Predictors of Explosive Synchronization with Ordinal Methods"

_entropy, 2025, doi:10.3390/e27020113_

Round 1

Reviewer 1 Report

Comments and Suggestions for Authors

The author proposes using the ordinal pattern transition (OPT) entropy measured at the sensor to predict the explosive transition in networks with dynamic systems. It shows a numerical analysis of different systems, including applying time series data from a network of electronic circuits operating in a chaotic regime.

This proposal is a starting point for future research focusing on refining ordinal measures to enhance their sensitivity and applicability to more extensive and diverse networks. Additionally, integrating machine learning with ordinal methods may prove beneficial in extending their predictive capabilities across various fields such as neuroscience, ecology, and engineering. 

The authors should consider publishing the raw data for further discussion and comparison of results.

Author Response

Comments 1: The author proposes using the ordinal pattern transition (OPT) entropy measured at the sensor to predict the explosive transition in networks with dynamic systems. It shows a numerical analysis of different systems, including applying time series data from a network of electronic circuits operating in a chaotic regime. 

This proposal is a starting point for future research focusing on refining ordinal measures to enhance their sensitivity and applicability to more extensive and diverse networks. Additionally, integrating machine learning with ordinal methods may prove beneficial in extending their predictive capabilities across various fields such as neuroscience, ecology, and engineering.  

The authors should consider publishing the raw data for further discussion and comparison of results. 

Response 1: We thank this Referee for taking the time to review our manuscript and for the positive assessment.   

We acknowledge the reviewer’s suggestion to publish the raw data to favor reproducibility and further research.  

Modification: We have added the following statement in Section Results III. D:  “... and the dataset can be downloaded from [71]” and added the corresponding reference: 

 [71] J. R. Sevilla Escoboza, I. Leyva Calleja, I. Sendiña Nadal, and J. Martin Buldú, Dataset on: “Explosive first-order transition to synchrony in networked chaotic oscillators.”, 10.5281/zenodo.14146078 (2024). 

Reviewer 2 Report

Comments and Suggestions for Authors

The authors have presented an interesting study for the use of ordinal methods to predict explosive transitions. The work is well written and supported by several computational investigations. I only have a few minor questions, to help enhance the clarity and reproducibility of the work.

- Please specify whether your source of stocasticity is dynamic noise (as normally in studies for EWS; if so, specify its properties and how it enters the equations) or parameteric uncertainty (as it appears from \omega_l,o); in the latter case, it would be useful to make it more explicit and briefly discuss its significance and what would someone expect by the theory.

- It is not clear how the order parameters d are chaged in the experiments: do they evolve gradually (e.g. in linear fashion) or are they at quasy-steady state for each simulation?

- The choice of varying d has an effect on the interpretation of the results. In fact, gradually changing parameters may have an effect on the computed statistics (for instance, the paper "Systematic analysis and optimization of early warning signals for critical transitions using distribution data" observes that a drop and relapse in AC1, very similar to Fig. 1, bottom, can be exmplained from commensurable time scales). Alternatively, one may try to derive analytical results to help guide the interpretation of the results. In general, such interpretation is what is less clear in the article. Focusing on the Kuramoto case, it is unclear how you define "EWS". Let's get back to AC1: one notices a drop before the transition. By the theory, the EWS is only the final increase just before the transition. Here, the EWS for the H measures seems to be more the drastic final drop than the wiggling increases. How can you tell one apart from the other? Also for the std: how can there be desynch if you're not going forward? It would thus look like std is an even earlier predictor. Trying to disentangle these effects, beyond eyeball visualization and expectations, would be significantly useful to better interpret the results. In general, is there a theretical benchmark to help us understand whatis going on? Or some theoretical contribution, to shed more light?

- Using permutation entropy for epileptic seizures has a decade-long history, but not many successes. Since you refer to epilepsy several times in the paper, you may be interested in discussing this issue, and how your study may contribute more in practice to advance the application of such measure in real-world settings.

- Related to this last question (but maybe beyond the scopes of this article and left for future investigations): entropy was observed to be a reliable indicator in other settings (see again the paper "Systematic analysis and optimization of early warning signals for critical transitions using distribution data") even in case of multiplicative noise. Other studies (e.g. "Warning signs for non-Markovian bifurcations: colour blindness and scaling laws") have also considered that colored noise changes the reliability of other EWS: do ordinal methods address those issues?

Author Response

Comments 1: T The authors have presented an interesting study for the use of ordinal methods to predict explosive transitions. The work is well written and supported by several computational investigations. I only have a few minor questions, to help enhance the clarity and reproducibility of the work. 

Response 1: We thank the Referee for taking the time to review our manuscript and for the positive assessment. We expect to have addressed all the concerns raised to improve the quality of our manuscript. 

Comments 2: Please specify whether your source of stochasticity is dynamic noise (as generally in studies for EWS; if so, specify its properties and how it enters the equations) or parameter uncertainty (as it appears from \omega_l,o); in the latter case, it would be useful to make it more explicit and briefly discuss its significance and what would someone expect by the theory. 

Response 2:  We thank the reviewer for pointing out this important issue. In the numerical parts of the work, the only source of stochasticity comes from the parametric uncertainty added to the leaves of the star network configuration to avoid a persistent synchronous state when reducing the coupling strength during the backward route to desynchronization. This parametric uncertainty is minimal and produces slight variations of the critical coupling strength for the explosive synchronization when performing different simulation instances and preserves the frequency gap between the hub and the leaves. Otherwise, the transition to synchronization is no longer abrupt. However, notice that, as the nodes are embedded in a network, the inputs from their neighborhood act as an incoherent source in each node's dynamics, playing the role of stochasticity. However, as the coupling increases, the properties of this input change and, therefore, cannot be characterized as done in Ref. 11 mentioned by the reviewer in the next comments.  

In our paper, the other stochastic source comes from the unavoidable tolerances of the electronic component parameters and the noise from uncontrolled sources in the experimental setup (Section III.D). However, those sources are not related to the transition itself.  

Modification: We have added a more detailed description of these stochasticity sources in the Methods section and briefly discussed their significance in the revised version of the manuscript. 

Comments 3: It is not clear how the order parameters d are changed in the experiments: do they evolve gradually (e.g., in a linear fashion), or are they at a quasi-steady state for each simulation?  

Response 3:  

The variation of the coupling strength d acting as a control parameter is NOT gradual during the evolution of the dynamical network. The system evolves for a long transient for each parameter value until the phase order parameter R reaches a stationary value. After that, we increase the value of d, starting the simulation, taking as initial conditions for the phases the ones left in the previous state. Therefore, each point in our simulations comes from a stationary data series.  

Modification: We have rearranged the Methods section and added a couple of paragraphs to clarify this point and how the simulations were conducted. 

Comments 4: The choice of varying d has an effect on the interpretation of the results. In fact, gradually changing parameters may have an effect on the computed statistics (for instance, the paper "Systematic analysis and optimization of early warning signals for critical transitions using distribution data" observes that a drop and relapse in AC1, very similar to Fig. 1, bottom, can be explained from commensurable time scales). Alternatively, one may try to derive analytical results to help guide the interpretation of the results. In general, such interpretation is what is less clear in the article. Focusing on the Kuramoto case, it is unclear how you define "EWS". Let's get back to AC1: one notices a drop before the transition. By the theory, the EWS is only the final increase just before the transition. 

Here, the EWS for the H measures seems to be more the drastic final drop than the wiggling increases. How can you tell one apart from the other? Also for the std: how can there be desynch if you're not going forward? It would thus look like std is an even earlier predictor. Trying to disentangle these effects, beyond eyeball visualization and expectations, would be significantly useful to better interpret the results. In general, is there a theretical benchmark to help us understand what is going on? Or some theoretical contribution, to shed more light? 

Response 4: We thank the Referee for pointing out this issue. As clarified in the previous comment, we chose to evolve the system for a fixed value of the control parameter d until a stationary state is reached. However, we also performed simulations by gradually (linearly) increasing the coupling strength d and observed similar results in all the indicators (not shown in the manuscript). 

The explosive synchronization (ES) in complex networks is characterized by the fact that the usual global observables (phase synchronization or absolute synchronization error) are mainly blind to the proximity of the transition. As to our knowledge, the phenomenon had not been studied from the point of view of transition prediction from signals, and therefore, in the manuscript, we intended to test if known measures used in the prediction of tipping points, as studied in Ref 11 suggested by the Referee, were any useful in our systems undergoing ES, including the proposal of OPT entropy as a possible addition to the prediction tools.  

Regarding how we define the EWS, we propose the relative difference observed between the ordinal pattern transition (OPT) entropy of the most connected node in the network and the less connected one, which increases much before the critical transition point (from d~0.015 in advance, Fig. 1).  In approximately this same range, AC1 decreases.  Regarding the drastic final relapse of hub AC1, which also coincides with the final drop of the OPT entropies, we do not consider them EWS as they happen simultaneously to the transition, recovering the value for the uncoupled system d=0 as expected. We apologize if the resolution in Fig. 1 is not clear enough in this sense; we have added a modification to clarify this.  Our purpose was to show how, during the long regime of very low values of the phase order parameter before the abrupt change to the synchronous phase, the hub's OTP entropy senses microscopical signals of changing dynamical organization, as it has been the case in all the shown examples.  

In our case, the time scales involved in the hub and the leaves are set such that the frequency gap is large enough to keep their natural frequencies until they become locked at the transition point. We have tried several cases for the node frequencies and/or the data sampling, always obtaining similar results. Therefore, it is improbable that this is due to commensurable time scales. However, we keep in mind that the hypothesis needs further research.  

About the theoretical insight and the comparation with the theory in Ref. 11, the referee is right in considering that it will be helpful. However, it becomes complicated because the observable used as input (the instantaneous frequencies of individual nodes in the case of Kuramoto or the local Poincaré sections for Rossler and Chialvo maps) is not the same one that undergoes ES, the global phase synchronization value R.  Therefore, the interesting theoretical considerations in Ref. 11, or even the more specific framework for ES found in Sci. Adv. 7, eabe3824 (2021) cannot be directly applied to our situation. In extended networked systems, it is more common to have access to local measurements rather than global synchronization data. In addition, as previously said, the system is not properly stochastic, as in Ref. 11, since the perturbation of the nodes' dynamics comes from the network coupling.  This is why our work could be interesting, even though it takes a phenomenological approach. 

We thank the reviewer for the insight that we acknowledged in the revised version.    

We believe we were not sufficiently clear regarding the behavior of the std. In all the figures, we show two synchronization curves for the phase order parameter R: the forward (black solid line) and backward (dashed line) continuations of the control parameter d. The forward curve is performed by gradually increasing the value of d and computing the stationary value of R. In contrast, the backward curve is performed by decreasing progressively d starting from a stable synchronous state.   

As we intend to detect the rising transition from incoherence to coherence, all measurements (OPT entropy, AC1, and std) are performed only for the forward continuation.  The backward curve is plotted only to frame the hysteretic region and, in this way, to better evidence the surprising fact that std seems to alert about the beginning of the hysteretic region, even if subsequently decreases to slightly rise again, to drop simultaneously to the transition. Additionally, the hub’s OPT entropy only diverges from the leaf one in this region. The large bump in std, even if it has dynamical justification, could be considered a false alarm from the ES's point of view.  

Modification: We have added several paragraphs along the manuscript to clarify these points 

Comments 5: Using permutation entropy for epileptic seizures has a decade-long history, but not many successes. Since you refer to epilepsy several times in the paper, you may be interested in discussing this issue, and how your study may contribute more in practice to advance the application of such measure in real-world settings. 

Response 5: We thank the reviewer for this suggestion, which motivated us to add a short discussion about this topic. Indeed, many papers report the use of permutation ordinal methods for epileptic seizures [72-75], but, as far as we know, we did not find any work using ordinal pattern transition entropies of spatially distributed EEG signals.  

Modification: We have added a short paragraph in the Conclusions section stating that future research should apply ordinal patterns transition entropy to predict the onset of epileptic seizures. 

Comments 6: Related to this last question (but maybe beyond the scopes of this article and left for future investigations): entropy was observed to be a reliable indicator in other settings (see again the paper "Systematic analysis and optimization of early warning signals for critical transitions using distribution data") even in case of multiplicative noise. Other studies (e.g., "Warning signs for non-Markovian bifurcations: colour blindness and scaling laws") have also considered that colored noise changes the reliability of other EWS: do ordinal methods address those issues?  

Response 6: We appreciate the reviewer’s comment regarding the robustness of the permutation entropy as an EWS in noisy environments. One of the advantages of the permutation entropy introduced by Bandt and Pompe [50] is that it works well with any real-world time series and is robust in the presence of observational and dynamical noise. An interesting question for future studies will be precisely to study the sensitivity to the transition’s proximity of local measures of ordinal transition entropies.  

Modification: We have added a comment in the Conclusions section regarding the robustness of ordinal transition methods to predict abrupt transitions. The two references suggested by the reviewer are cited in the Introduction as Refs. [11] and [12] and throughout the manuscript in the revised version.